# Methylglyoxal: A Key Factor for Diabetic Retinopathy and Its Effects on Retinal Damage

**DOI:** 10.3390/biomedicines12112512

**Published:** 2024-11-02

**Authors:** Vladlen Klochkov, Chi-Ming Chan, Wan-Wan Lin

**Affiliations:** 1Graduate Institute of Medical Sciences, Taipei Medical University, Taipei 11031, Taiwan; klochkovvladlen@gmail.com; 2Department of Ophthalmology, Cardinal Tien Hospital, New Taipei City 23148, Taiwan; 3School of Medicine, Fu Jen Catholic University, New Taipei City 242062, Taiwan; 4Department of Pharmacology, College of Medicine, National Taiwan University, Taipei 100233, Taiwan

**Keywords:** methylglyoxal, diabetic retinopathy, oxidative stress, inflammation, autophagy, ER stress, glyoxalase

## Abstract

**Background:** Diabetic retinopathy is the most common retinal vascular disease, affecting the retina’s blood vessels and causing chronic inflammation, oxidative stress, and, ultimately, vision loss. Diabetes-induced elevated glucose levels increase glycolysis, the main methylglyoxal (MGO) formation pathway. MGO is a highly reactive dicarbonyl and the most rapid glycation compound to form endogenous advanced glycation end products (AGEs). MGO can act both intra- and extracellularly by glycating molecules and activating the receptor for AGEs (RAGE) pathway. **Conclusions**: This review summarizes the sources of MGO formation and its actions on various cell pathways in retinal cells such as oxidative stress, glycation, autophagy, ER stress, and mitochondrial dysfunction. Finally, the detoxification of MGO by glyoxalases is discussed.

## 1. Sources of Methylglyoxal and Its Association with Diabetic Retinopathy

Diabetes mellitus (DM) is one of the most widespread diseases today. According to the International Diabetes Federation, 1 in 10 people suffers from this condition [1]. Elevated blood glucose levels cause dysfunction in various tissues, reducing people’s quality of life.

The accumulation of advanced glycation end products (AGEs) during DM’s progression is associated with biochemical tissue dysfunctions, such as nephropathy, retinopathy, and peripheral neuropathy [2,3,4]. Diabetic retinopathy (DR) results from chronic DM and is the most common retinal vascular disease. Statistics show that three out of four people who have lived with DM for more than 15 years develop DR [5]. DM is a complex disease that has several stages of progression. First, it starts with mild non-proliferative DR with the appearance of small microaneurysms. The second stage (moderate non-proliferative DR) is characterized by further microaneurysm progression, bleeding, and deposits. The third stage (severe non-proliferative DR) promotes blood vessel abnormalities. After that, in the fourth stage (proliferative DR), new abnormal blood vessel formation (neovascularization) is induced and leads to retinal detachment and vision loss. Currently, DR is the fifth leading cause of vision impairment. Elevated levels of glucose, blood pressure, and glycated proteins are significantly associated with the progression of DR [6]. Common methods to study the progression of DR in animals include streptozotocin- and alloxan-induced models, high-fat diets (HFD), and genetic approaches to diabetes progression [7,8]. Another potential way to study DR is through the application of AGEs and methylglyoxal (MGO) [9].

MGO is a highly reactive dicarbonyl compound that plays a significant role in the pathogenesis of DM, its metabolic complications, and other age-related chronic inflammatory diseases such as cardiovascular diseases and cancer [10,11,12,13,14]. MGO is formed as a byproduct of glycolysis and is the primary and fastest cause that is known to non-enzymatically modify protein and DNA by glycation and lead to AGE formation [11,15].

Until now, several pathways of MGO formation have been identified. The major source coming from glucose is via the polyol pathway, where aldose reductase causes the reduction of glucose to sorbitol, and sorbitol dehydrogenase converts sorbitol to fructose [16]. Aldolase then converts fructose to dihydroxyacetone phosphate (DHAP). The activities of both aldose reductase and sorbitol dehydrogenase are increased in DM conditions [17]. MGO synthase has been found to convert dihydroxyacetone phosphate into MGO irreversibly. Under DM conditions, DHAP levels are significantly elevated to increase MGO formation [18].

MGO can also be produced from the metabolism of proteins and fatty acids, although only in small amounts [19]. The increased formation of ketones, such as acetone, found in DM can activate CYP2E1 (an acetone monoxidase) in the liver to form MGO. The formation of MGO from proteins is catalyzed by semicarbazide-sensitive amine oxidase (SSAO), which converts aminoacetone to acetone, acetol, and, finally, to MGO [20]. Figure 1 shows a schematic representation of the major sources of MGO production.

In addition to the generation of MGO from the oxidation of carbohydrates, lipids, and proteins, as mentioned above, in various tissues, vitamin A-derived lipofuscin pigments bisretinoids are also a unique and novel source of MGO in the eye [21,22]. Upon irradiation with short-wavelength visible light, the bisretinoid fluorophores such as A2E and all-trans-retinal dimers are formed as byproducts of vitamin A cycling in the retina and accumulate in retinal pigment epithelial (RPE) cells. Further photooxidation and photodegradation of both bisretinoids lead to the formation of MGO. In this review, we focused on the pathogenetic role of MGO in DR, and the molecular mechanisms of its actions in retinal cells, particularly in RPE, Müller, endothelial, and microglial cells.

## 2. Modes of Action of MGO in DR

MGO has been shown to affect numerous cellular targets, signaling pathways, and stress responses in retinal cells, leading to impaired retinal function. Two major action modes of MGO, i.e., direct non-enzymatic glycation and receptor for AGEs (RAGE) pathway activation, are integrated and contribute to its pathogenic actions.

### 2.1. MGO-Induced Retina Protein Glycation

Reactive dicarbonyls play a crucial role in protein glycation due to their high reactivity. Glycation adducts, such as hydroimidazolone AGEs, are produced from MGO and are the major AGEs formed in the retina, nerves, glomeruli, and plasma proteins [3]. The high levels of protein glycation via the AGEs/MGO pathway, especially Nε-carboxymethyl lysine adducts in the serum, vitreous, and retina are correlated with the progression and various secondary complications of DM [23,24,25]. These complications include neuropathy of the retina [2,26,27,28,29], nephropathy [30,31], impaired wound healing [32], cardiovascular complications [12,24,33,34], salivary protein glycation [35], and neurodegeneration [27]. MGO is also suggested to be involved in insulin resistance and beta-cell dysfunction, contributing to the development of DM and creating a feedback loop between glycation and hyperglycemia [24,30]. Protein glycation in the retina contributes to vision loss during DR progression. The accumulation of glycated proteins in the lens causes blurred vision in the early stages of DR and may lead to complete vision loss in the advanced stages [36].

In addition to the well-known glycated hemoglobin (HbA1c), which is a disease progression marker of DM, there are several identified glycated proteins of AGEs that contribute to abnormalities in retinal cell pathways, and cytotoxicity [37]. These include crystallins, albumin, low-density lipoprotein (LDL), and extracellular matrix. Water-soluble lens proteins, such as crystallins, are crucial for maintaining retinal transparency and participating in metabolic and regulatory functions [38]. Abdullah et al. found that MGO-induced glycation of camels’ lens ζ-crystallin alters its secondary structure and reduces its solubility. They also found that MGO can glycate proteins more rapidly than high glucose levels [39]. Moreover, α,β-crystallin in RPE cells exhibits antiapoptotic activity against MGO treatment. The interaction between α,β-crystallin and caspase subtypes −2L, −2S, −3, −4, −7, −8, −9, and −12 can be disrupted by MGO treatment, causing caspase release and cellular apoptosis [40]. In addition, the albumin glycation that forms under hyperglycemic conditions in the retinal blood vessels of diabetic patients is involved in DR. Glycated albumin can induce cell death in retinal pericytes via reactive oxygen species (ROS) production [41], retinal RPE dysfunction [42], vascular injury [43], blood–retinal barrier (BRB) permeabilization [44], and inflammation in the retinal microglia [45]. Glycated LDL may mediate capillary injury in DR [46]. Moreover, glycated extracellular matrix proteins, such as fibronectin and laminin, are detected in diabetic animals [47], and in endothelial and Müller cells in the retinas of patients with DR [48].

Oxidative stress is the most important key contributor to the pathogenesis of DR and AGE-induced retinal injury [49]. Accumulating lines of evidence have demonstrated that AGEs act as pro-oxidant metabolites, which lead to elevation of intracellular ROS and lipoxidation [50]. All these studies suggest that antioxidants and inhibitors of advanced glycation such as Nrf2 activators and pigment epithelium-derived factor (PEDF) are therapeutic strategies to ameliorate DR [44,51,52].

### 2.2. MGO-Induced RAGE Activation

MGO’s mode of action is not limited to protein glycation but also involves the activation of RAGE. The AGEs–RAGE axis can trigger a range of signaling events that are associated with diabetes [53,54]. The activation of RAGE in RPE and Müller cells induces the NF-κB, PI3K/AKT/GSK3β, Ras/MEK/ERK, p38, and JNK pathways and increases nicotinamide adenine dinucleotide phosphate (NADPH) oxidase activity [55,56]. Moreover, activation of RAGE by AGEs also mediates cellular dysfunction and apoptosis in RPE cells [13,57]. As well as in the retina, the involvement of the MGO–RAGE axis in multiple pathological consequences related to neurodegeneration has been demonstrated in the brain, where RAGE activation mediates disruption of the blood–brain barrier, neuroinflammation, remodeling of the extracellular matrix, and dysregulation of the polyol pathway and antioxidant enzymes [58].

Studies have also shown the positive feedback regulation of the AGEs–RAGE pathway and the interplay between RAGE and TLR4 in the pathogenesis of DR. In a diabetic mouse model, it was found that diabetes progression and MGO accumulation can activate and upregulate RAGE’s expression in the retina [59]. This event might result from the ROS-NF-κB axis [60,61]. Ramya et al. found that MGO and its induced AGEs promote inflammation and expression of TLR4 in endothelial cells via the RAGE pathway [62]. Moreover, high mobility group box 1 (HMGB1) is an inflammatory alarmin that initiates the host’s defense system. Studies have indicated that the role of HMGB1 in the pathogenesis of Type 2 DM depends on the activation of the RAGE and TLR4 molecules that contribute to the production of pro-inflammatory cytokines [63,64]. Moreover, AGEs can also increase HMGB1 secretion in retinal ganglion cells, which, in turn, activates RAGE for vascular endothelial growth factor (VEGF)-A production [65]. With these findings, manipulation of RAGE’s activity by either reducing RAGE’s expression [66] or antagonizing RAGE [67,68] becomes a promising strategy to prevent AGE-induced retinal cell damage and the progression of diabetic complications.

## 3. MGO Alters Various Cellular Pathways

### 3.1. Inhibition of Autophagy in RPE Cells

Autophagy is a catabolic process that degrades and recycles damaged organelles, cellular components, and cytoplasmic proteins to maintain cellular homeostasis. The primary function of RPE cells is to maintain healthy photoreceptors [69]; however, their functions also include light absorption, phagocytosis, barrier function, and participation in the retinoid cycle. RPE cell dysfunction can be caused by oxidative stress [70,71], senescence [72,73], Type 2 DM [74,75], and MGO [2,9,76,77]. Impaired RPE function promotes photoreceptor damage and other cellular injuries, increases chronic inflammation, and ultimately leads to vision loss [78]. RPE cells exhibit high levels of autophagy proteins to maintain their phagocytic functions [79,80]. Normal autophagy levels prevent RPE cells’ dysfunction, as demonstrated in studies on MGO-, UVA-, and NaIO_3_-induced damage [76,81,82]. Abnormal levels of autophagy, either decreased [82] or increased [83], promote cellular damage. MGO-modified proteins inhibit autophagy, and impaired autophagy promotes retinal cell death and inflammation via the NLRP3 inflammasome pathway [84].

In models of high glucose and diabetes, decreased phagocytosis and autophagy levels are linked to the cellular signaling interplay among the AMPK, AKT, and mTOR pathways. High-glucose treatment of ARPE-19 cells reduces AMPK but increases AKT activities, downregulating mTOR-related autophagy genes’ expression [85,86]. Feng et al. demonstrated the protective role of autophagy in DR. RPE cells exposed to high glucose showed decreased autophagy levels and lysosomal membrane permeabilization, which was abolished by HMGB1 silencing-dependent restoration of the degradative capacity of autophagy [87]. In our previous work, we found that MGO treatment causes significant apoptosis in ARPE-19 cells [76]. MGO treatment affects autophagy proteins similarly to high glucose, downregulating the LC3II/LC3I ratio via suppression of AMPK, suggesting that MGO-induced downregulation of autophagy in RPE cells promotes cell death.

### 3.2. Induction of Oxidative Stress-Associated Inflammation in Retinal Cells

Inflammation induced by various stress factors, along with related oxidative stress, leads to cellular dysfunction. High glucose levels have been shown to promote oxidative stress and inflammation in RPE cells [88,89], and the production of both ROS and/or reactive nitrogen species (RNS) can be induced by high glucose in RPE cells [90,91], retinal ganglion cells [92], photoreceptor cells [93], and retinal microvascular endothelial cells [94]. Likewise, MGO can increase the production of ROS and RNS, damaging cellular components, leading to mitochondrial dysfunction, cell death, and inflammation in various cell types, including PREs [76,95], retinal pericytes [96], endothelial cells [97], eosinophils [98], macrophages [99], and hepatocytes [100].

MGO has also been found to induce inflammation responses in the retina of humans and rats with DM. MGO treatment elevates the expression of cyclooxygenase (COX)-2, chemokine receptor CXCR4, IL-6, IL-8, monocyte chemoattractant protein-1 (MCP-1), and intercellular adhesion molecule 1 (ICAM-1) genes [101]. MGO-modified fibronectin was shown to upregulate CD40, ICAM-1, and CCL2 in endothelial and Müller cells, leading to enhanced CD40-dependent pro-inflammatory responses. Moreover, increased CD40 expression in endothelial and Müller cells from patients with DR was observed by confocal microscopy, which was associated with increased carboxymethyl lysine expression in fibronectin and laminin [28]. In human retinal endothelial cells, MGO also can upregulate the expression of lysyl oxidase via RAGE, leading to pro-inflammatory and matrix stiffening [102]. Recently, Wang et al. reported that MGO can induce pyroptosis in endothelial cells via NLRP3 inflammasome activation and oxidative stress [97]. Moreover, MGO displays the ability to reduce the immunosuppressive activity of retinal pericytes to inhibit activated T cell proliferation via expressing PD-L1. This finding reveals that the protective function of pericytes against inflammation-mediated apoptosis in the retina is impaired by hyperglycemic conditions [103]. As mentioned above, MGO-induced inflammation might result from the direct activation of RAGE and the consequential activation of TLR [62,98].

### 3.3. Inflammation and Microglia Activation

Müller cells are the principal macroglial cells of the retina. Their main functions include providing structural support (by maintaining the structural integrity of the retinal layers), regulating the retinal environment (by balancing ions, removing excess neurotransmitters such as glutamate, and maintaining K^+^ concentrations in the retinal tissue), light conduction, nutrient transport, and neuroprotection (by secreting neurotrophic factors) [104]. Microglia are the resident immune cells of the CNS, including the retina, and play a critical role in responding to injury, infection, and other pathological changes while maintaining retinal homeostasis [105]. Studies have indicated that Müller cells and microglia work together to maintain retinal homeostasis, with Müller cells playing a role in regulating microglial activation and function [106]. This cell–cell interaction is crucial for maintaining a balanced immune response in the retina [107]. However, in DR, the microglia become overactivated and contribute to disease progression. The inflammatory response induced by activated microglia can damage the BRB, leading to increased vascular permeability [108], which contributes to retinal edema and hemorrhage, the hallmark features of DR [109]. Persistent activation of the microglia in the diabetic retina leads to a chronic inflammatory state, exacerbating retinal damage and accelerating the progression of DR from the non-proliferative to the proliferative stage.

Reber et al. demonstrated that glyoxal causes significant morphological changes in the E1A-NR3 retinal neuron cell line, including cell membrane blebbing, aggregation of intracellular organelles, time-dependent acidification (to pH 7.2), loss of mitochondrial membrane potential (MMP), and apoptosis [109]. MGO and high-glucose conditions activate Müller cells’ CD40 receptor and produce pro-inflammatory cytokines (TNF-α, IL-1β, CCL2) via upregulation of the CD40/MAPK/NF-κB pathway [48,110]. Portillo et al. further highlighted the role of stress-induced activation of Müller cells and microglia (Figure 2). CD40 activation not only promotes cytokine synthesis but also stimulates the TRAF2/Src/PLCγ1 pathway, leading to ATP release from Müller cells, the activation of microglial P2X7 receptors, and their chronic activation [111,112,113].

The formation of microaneurysms, hemorrhages, and areas of necrosis (cotton wool spots) promotes retinal hypoxia and neovascularization. Retinal neovascularization in the last stages of DR can be promoted through several molecular pathways. DR-induced hypoxia increases the production and stability of hypoxia inducible factor 1-α (HIF-1α) protein. HIF-1α stimulates VEGF and nitric oxide production [114]. VEGF is an angiogenic factor associated with abnormal blood vessel formation and permeability, mostly produced by activated Müller cells in the retina [115]. VEGFR activation promotes inflammation (via MAPK), angiogenesis, and blood leakage (via PI3K/AKT). In a zebrafish embryo model of MGO-induced retinal angiogenesis, MGO was found to promote angiogenesis through VEGF production [116]. These MGO-induced changes were reversed by the VEGF inhibitor PTK787, proving the important role of VEGF synthesis under MGO treatment. Bautista-Pérez et al. further confirmed this hypothesis in a streptozotocin-induced diabetic rat model [25]. They showed that high glucose levels in diabetic conditions lead to a pro-inflammatory state in the retinal cells via the RAGE/NF-κB pathway, which is also linked to the production of VEGF and the progression of proliferative DR. On the other hand, nitric oxide induces vasodilation and increases VEGF production via the PI3K/AKT/HIF-1α pathway, forming a loop that potentiates DR progression [117]. Another molecular pathway involved in DR’s neovascularization is angiopoietin (Ang). Ang1 can stabilize new blood vessels and the BRB’s integrity via the PI3K/AKT/FOXO1 pathway, while Ang2 has a destabilization effect [118]. Through DR progression, Ang1 levels are decreased while Ang2 is upregulated, resulting in DR complications [119]. In the MGO treated ARPE-19 cells and animal DR models, MGO was found to increase Ang2 levels and decrease the secreted VEGF/Ang 2 ratio in retinal epithelial cells, leading to the promotion of apoptosis, decreased endothelial cell proliferation, and increased microvascular permeability [120].

### 3.4. Endoplasmic Reticulum Stress and Ca^2+^ Signaling

The endoplasmic reticulum (ER) is an intracellular Ca^2+^ storage organelle that plays a crucial role in signal transduction and protein folding. ER stress primarily triggers three major signaling pathways: IRE1α/XBP1, PERK/eIF2α/ATF4, and ATF6. It can also lead to apoptosis and arrested cell growth via the CHOP protein [121,122]. ER stress plays a significant role in the progression of DR [101]. In our previous work, we found that MGO induces RPE cell death through ER stress-related ROS production and mitochondrial dysfunction [78]. MGO triggers both necrotic and apoptotic cell death modes. MGO increases the expression of proteins such as GRP78, CHOP, ATF6, and ATF4, along with phosphorylation of eIF2α and PERK, and induces spliced XBP1 and ATF6 formation in a time-dependent manner. In the first 2 h following MGO treatment, cellular and mitochondrial ROS levels increase. N-acetylcysteine can protect RPE cells from MGO-induced cell death. Additionally, 4-PBA, salubrinal (ER stress inhibitors), and BAPTA/AM (a Ca^2+^ chelator) can also prevent cell death, suggesting the involvement of ER stress and Ca^2+^ signaling [78]. Furthermore, ER stress inhibitors can reduce ROS production, prevent increases in intracellular Ca^2+^ levels, and protect against MMP loss. Additionally, the store-operated Ca^2+^ entry (SOCE) inhibitors MRS1845 and YM-58483, but not the IP_3_ receptor inhibitor xestospongin C, were able to block MGO-induced ROS production, MMP loss, and the sustained increase in intracellular Ca^2+^ in RPE cells [78]. These findings suggest that MGO is capable of inducing an ER stress response, leading to Ca^2+^ signaling changes, mitochondrial dysfunction, and cell death.

### 3.5. AMP-Activated Protein Kinase and Mitochondrial Stability

AMP-activated protein kinase (AMPK) is one of the key proteins involved in metabolism and energy homeostasis [123]. Impaired glucose metabolism in diabetic conditions leads to reduced glucose uptake and an altered ATP/ADP ratio, which downregulates AMPK activity [124]. It is also known that AMPK activation induces antioxidant responses through the PGC-1α, FOXO, and Nrf2 pathways. The interplay between SIRT1 and AMPK promotes antioxidant responses in diabetic conditions [125]. AMPK also plays a role in lipid metabolism. Reduced AMPK activity is implicated in lipid dysregulation, including dysregulated de novo lipogenesis, elevated acetyl-CoA carboxylase signaling, and increased fatty acid synthase activity, all of which contribute to lipotoxicity [126].

AMPK activation also promotes autophagy in RPE cells. MGO inhibits AMPK’s activity and autophagy (indicated by accumulation of LC3II) while decreasing the expression of mitochondrial biogenesis and dynamic markers (MFN1, PGC-1α, and TFAM) in RPE cells [95]. Both AMPK activators (metformin and A769662) reduce these effects of MGO. Moreover, Zou et al. found that Wnt inhibitory factor 1 (WIF1) can suppress the expression of VEGF, mitochondrial autophagy-related proteins, and mitochondrial dysfunction in high glucose-treated ARPE-19 cells. An in vivo study showed that WIF1 reduces DR by downregulating the AMPK/mTOR axis [86].

Additionally, AMPK activity is also involved in the upregulation of the Glo1 protein. Animal studies confirmed these findings, showing that an intravitreal injection of MGO causes cotton wool spots and macular edema. However, metformin and A769662 treatments in animal groups provide protective effects against MGO. Functional, histological, and optical coherence tomography analyses support the protective actions of AMPK activators against MGO-induced retinal damage [76]. Song et al. demonstrated AMPK’s protective effect in AGE-induced apoptosis in photoreceptor 661W cells and streptozotocin-induced photoreceptor cell degeneration [127]. MGO and AGE treatment of 661W cells induce apoptosis by promoting Bax but reducing Bcl-2 protein levels, as well as disrupting autophagy. AGE treatment also causes mitochondrial dysfunction, abnormal mitochondrial morphology; downregulation of mitochondrial biogenesis-associated proteins such as TFAM, Nrf1, and PGC-1α; and antioxidant protein expression. These changes in 661W cells are prevented by metformin pretreatment, suggesting AMPK’s protective role. The same effect has been confirmed in a diabetes-induced DR model in mice, which is also prevented by metformin administration [128]. A summary of the molecular pathways involved in MGO’s action in RPE cells is presented in Figure 3.

In Table 1, we summarize the retinal cell type-specific actions of MGO.

## 4. Role of Glyoxlases in MGO Detoxification

There are several detoxification systems for MGO in the body, including kidney clearance, aldehyde dehydrogenase, aldo-keto reductase, and the glyoxalase (Glo) pathways. The glyoxalase system is considered the most important for MGO detoxification in the retina (Figure 4). The first step of MGO detoxification is its reaction with glutathione for the formation of hemithioacetal. Glyoxalase 1 (Glo1) is a Zn^2+^-dependent enzyme that converts hemithioacetal to S-D-lactoylglutathione. Glo1 is regulated by proteins such as E2F4, NF-κB, Nrf2, and PKA [129]. Nrf2 acts as an inducer of Glo1, promoting its expression. PKA can phosphorylate Glo1 at Thr107, which decreases its activity and leads to caspase-dependent cell death [130].

Glyoxalase 2 (Glo2) exists in both the cytosolic and mitochondrial fractions, where it hydrolyzes S-D-lactoylglutathione to produce D-lactate and regenerates glutathione [131]. Glo2′s catalytic domain contains Fe^2+^ and Zn^2+^, but only Zn^2+^ can regulate its activity. Glo2 also has a p53-responsive region that is activated by p63 and p73, increasing its expression [132]. Lastly, the glyoxalase 3 (Glo3) protein, which converts MGO to D-lactate without the need for glutathione, is present in bacteria.

The detoxification of MGO via the Glo system is closely linked to complications of diabetes, including nephropathy, retinopathy, neuropathy, and cardiovascular disease [133]. Increased MGO formation in hyperglycemia is associated with the downregulation of Glo1 protein due to inflammatory signaling [134]. Glo1’s expression and activity are downregulated in diabetic patients [135]. It has been found that knocking out transient receptor potential cation channel (TRPC) 1/4/5/6 isoforms in mice increases Glo1’s activity and expression in diabetic conditions, protecting the retina from DR-related changes and reducing MGO formation [136].

Several studies on manipulating Glo1 levels in cell and animal models have demonstrated its crucial role against diabetes. In zebrafish, CRISPR-Cas9 knockout of Glo1 potentiates HFD-induced MGO formation, elevates fasting glucose levels, impairs glucose tolerance, and promotes new retinal blood vessel formation [137]. Bernel et al. showed that overexpression of Glo1 in streptozotocin-treated rats prevents DR-related retinal damage by reducing AGE formation, decreasing GFAP levels, increasing Kir4.1 protein expression in Müller cells, and reducing the formation of new blood vessels in the retina [138]. In Drosophila melanogaster, knockout of Glo1 accelerates diabetes progression, increases MGO concentrations, promotes lipid accumulation, elevates blood glucose levels, and decreases insulin sensitivity [139]. The Glo1 inducer tRES-HESP (a combination of trans-resveratrol and hesperetin) has been shown to reduce the expression of RAGE and cell adhesion molecules, and to decrease inflammation in human aortic endothelial cells [140]. In fibroblasts and HepG2 cells, tRES-HESP can reduce basal levels of RAGE and MMP3 proteins and increase glutathione levels [141].

## 5. Future Research Directions and Therapeutic Approaches

The retina is a complex tissue consisting of various cell types. The diversity of cells in the retina explains the insufficiency of data from single-cell-line models and the obligatory animal disease modeling. Coculturing and tissue engineering technologies might be a possible future key to resolving this problem in retinal research and research into MGO in particular [142]. Nowadays, several cocultured retinal cell models have been established: ARPE-19—endothelial cells [143], ARPE-19—microglia [144], ARPE-19—neurons [145], Müller cells—microglia [146,147], Müller cells—endothelial cells [148,149], and Müller cells—photoreceptors [150]. Furthermore, Achberger et al. investigated the human retina-on-a-chip platform for retinal neovascularization research [151]. Using these coculturing and organoid systems may shed light on MGO’s effects and also would be promising in DR drug discovery.

Future approaches in DR treatment should start with a better diagnosis of the disease. In this case, fascinating research on developing an AI-based deep learning tool for DR screening was carried out by Dai et al. [152]. Early-phase prediction and further treatment of DR promote the outcomes and prolong the healthy years of patients. As a promising new DR treatment, we can mention the Glo1 inducer tRES-HESP. Currently, it is the first Glo1 inducer that has finished a Phase 1 clinical trial [141] for improving glycemic control and vascular function in overweight and obese subjects. These data provide promising results but more detailed research into DR models needs to be carried out. Another possible way to induce Glo1 production is Nrf2 activation [153]. Several natural compounds found to activate Nrf2 abolish DR progression and retinal cell death [154,155,156]. RAGE inhibitors were found to be a prospective way to prevent DR-induced inflammation [68]. Li et al., in a mouse DR model, found that RAGE inhibitors inhibit capillary degeneration, nitration of retinal proteins, retinal leukostasis, and ICAM-1 expression, proving the potential of that treatment [67].

## 6. Conclusions

DR is one of the most common retinal vascular diseases promoting new blood vessel formation, chronic inflammation, and oxidative stress. Increased glucose levels and impaired lipid synthesis promote reactive dicarbonyl MGO formation. MGO is the most rapid glycation compound, which acts both intra- and extracellularly by non-enzymatically glycating proteins and activating the RAGE pathway, respectively. Its action causes several cellular responses that lead to retinal cell dysfunction and cell death. These include reduced autophagy, oxidative and ER stress, mitochondrial dysfunction, cellular calcium overload, AMPK inhibition, RAGE activation, and inflammation. In addition to supplementation with antioxidants to inhibit AGE formation and oxidative stress-mediated cell dysfunction, work towards discovering newer molecular targets of glyoxalase, glycation, RAGE, and aldose reductase for chronic DR is a new research field.

## Figures and Tables

**Figure 1 biomedicines-12-02512-f001:**
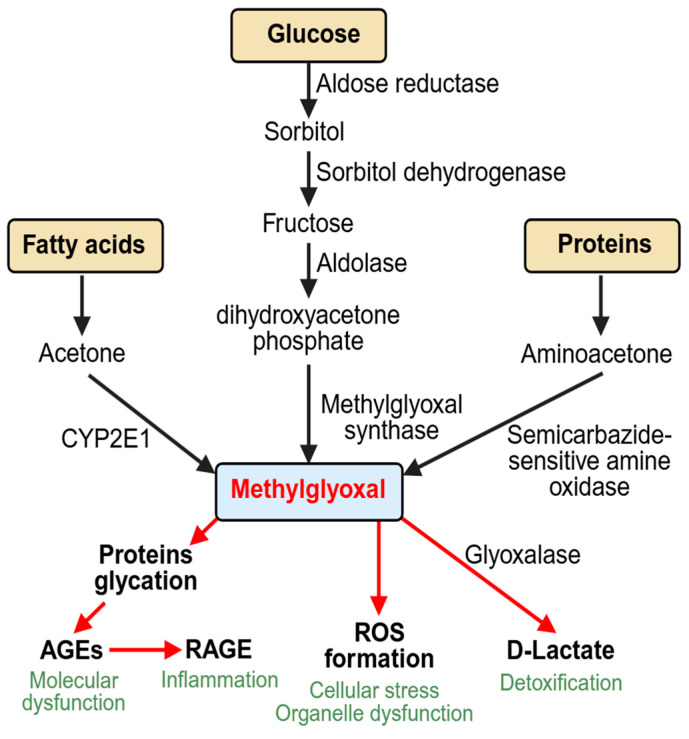
Schematic representation of methylglyoxal’s sources and cellular outcomes.

**Figure 2 biomedicines-12-02512-f002:**
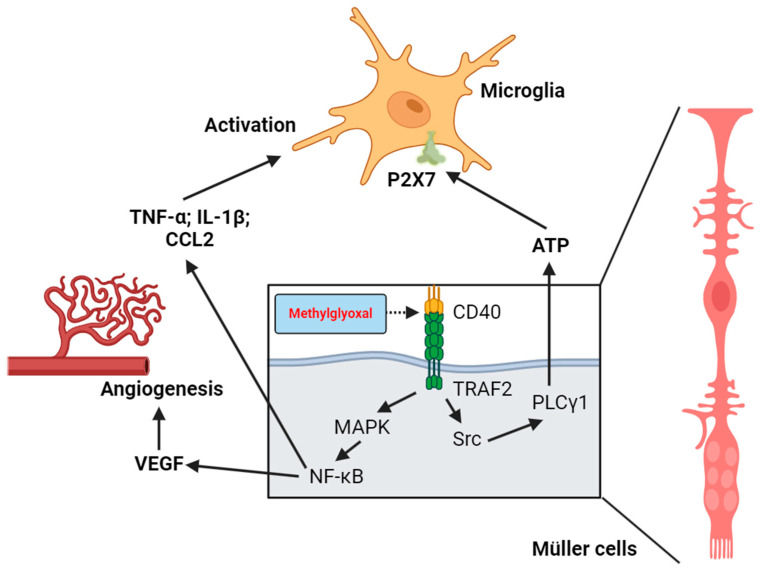
Roles of CD40 activation and Müller cell–microglia communication in MGO-induced chronic inflammation and promotion of new blood vessel formation in the retina.

**Figure 3 biomedicines-12-02512-f003:**
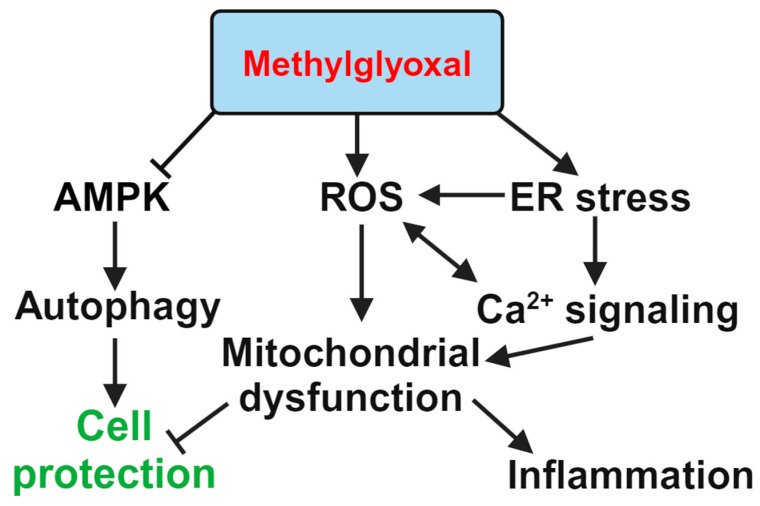
Representation of MGO-induced pathways in RPE cells.

**Figure 4 biomedicines-12-02512-f004:**
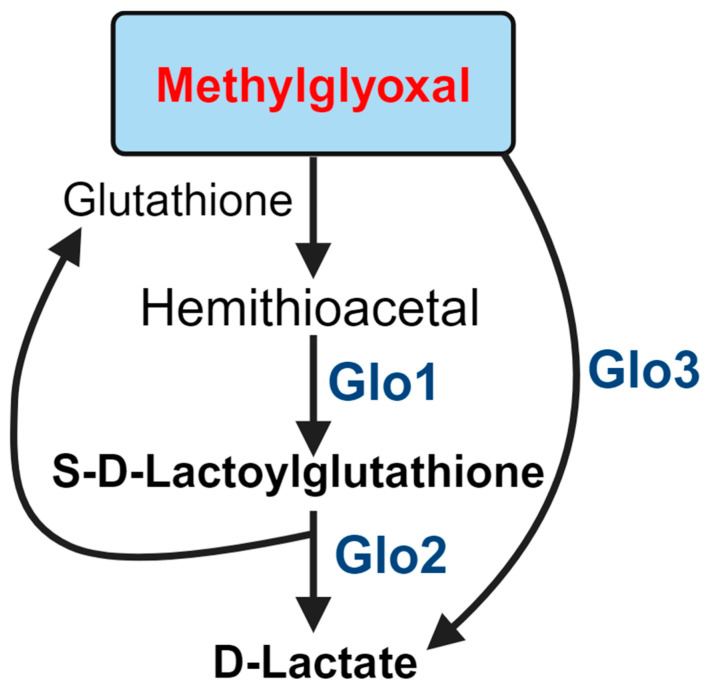
Role of the glyoxalase system in MGO detoxification.

**Table 1 biomedicines-12-02512-t001:** MGO’s effect on different retinal cell types.

Cell Type	MGO’s Effects	Reference
Retinal pigment epithelium cells	Promotes α, β -crystallin glycation, caspase release and apoptosis	Jeong et al. [40]
	Downregulates autophagy proteins via AMPK suppression, contributing to cell death	Sekar et al. [76]
	Induces ROS-induced mitochondrial dysfunction, inflammation, and cell death	Chang et al. [95]
	Induces ER stress-related ROS production and mitochondrial dysfunction	Chan et al. [7]
	Inhibits autophagy and mitochondrial biogenesis, and promotes mitochondrial fission via AMPK inhibition	Zou et al. [86]
Müller cells	Activates CD40 to induce the CD40/MAPK/NF-κB pathway and inflammation	Portillo et al. [48]
	Activates CD40 to induce the TRAF2/Src/PLCγ1 pathway, leading to ATP release and microglial P2X7 activation	Portillo et al. [110]
	Promotes VEGF production	Li et al. [116]
Retinal neuron cells	Induces morphological changes, MMP loss, and apoptosis.	Reber et al. [109]
Pericytes	Induces ROS-induced mitochondrial dysfunction, inflammation, and cell death	Kim et al. [96]
	Reduces immunosuppressive activity via PD-L1 expression	Tu et al. [103]
Retinal endothelial cells	Promotes ROS-induced mitochondrial dysfunction, inflammation, and NLRP3 activation via RAGE activation, resulting in pyroptosis.	Wang et al. [97]
	Upregulates ICAM-1 and CCL2 proteins, and promotes CD40-induced inflammation	Portillo et al. [48]
	Upregulates lysyl oxidase expression and inflammatory markers via RAGE activation	Chandrakumar et al. [102]

## Data Availability

Data used for this review are available online on PubMed.

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
