# Peer review of "Methylglyoxal: A Key Factor for Diabetic Retinopathy and Its Effects on Retinal Damage"

_biomedicines, 2024, doi:10.3390/biomedicines12112512_

Round 1

Reviewer 1 Report

Comments and Suggestions for Authors

The manuscript provides an extensive review of MGO and its role in the pathogenesis of DR. The paper first discusses MGO's formation through glycolysis and other metabolic pathways and its damaging effects on retinal cells via glycation and the activation of RAGE. It further addresses MGO's impact on oxidative stress, inflammation, autophagy, and mitochondrial dysfunction, and discusses detoxification mechanisms involving glyoxalases. This review is overall comprehensive but could benefit by adjusting some discussions and expanding the therapeutic and clinical perspectives, particularly in relation to drug development targeting MGO.

Comments:

1. Some abbreviations lack their full name upon their first appearance, such as RAGE, DHAP, BRB, etc.

2. Figure 1 caption needs to be modified because Figure 1 includes not only the sources of MGO, but also some downstream effects. Also, the location of “AGEs” and “Proteins glycation” may need to be switched because AGE should be one of the consequences of protein glycation.

3. How do glycated crystallins, albumin, LDL, and ECM lead to ROS and cytotoxicity via glycation? Are they through activation of RAGE, or other AGE receptors? If it is through RAGE, this part of the discussion should be moved to section 2.2; Otherwise, it would be helpful to have a new section 2.3 discussing other downstream receptors of AGEs apart from RAGE.

4. It is nice to have some discussion regarding the potential therapeutic application of glyoxalase to detoxify MGO. However, more discussion from a clinical perspective would be helpful for the reader to understand the current stage of drug development targeting MGO. Are there any relevant ongoing clinical trials? If not, what would be the major obstacles to drug development?

5. Retina is a complex tissue comprising multiple cell types, and they also tightly interact with each other. From the discussions in this manuscript, MGO majorly affects RPE, muller cells, photoreceptors, and vasculatures. It would be beneficial to briefly summarize the role of MGO on each cell type. Also, are there any tissue engineering models or in vivo models showing how MGO collectively affects them? The following reviews may provide useful context regarding the current retina coculture models:

Marcos LF, Wilson SL, Roach P. Tissue engineering of the retina: from organoids to microfluidic chips. J Tissue Eng. 2021 Dec 10;12:20417314211059876. doi: 10.1177/20417314211059876. PMID: 34917332; PMCID: PMC8669127.

Wu A, Lu R, Lee E. Tissue engineering in age-related macular degeneration: a mini-review. J Biol Eng. 2022 May 16;16(1):11. doi: 10.1186/s13036-022-00291-y. PMID: 35578246; PMCID: PMC9109377.

Author Response

Dear Reviewer,

We would like to thank referee for reviewing our manuscript and providing helpful comments. Please find the detailed responses below and the corresponding revisions highlighted changes in the re-submitted files.

Comment 1: Some abbreviations lack their full name upon their first appearance, such as RAGE, DHAP, BRB, etc.

Reply: Thank you for finding these inaccuracies, as you suggested we added the full names of abbreviations.

Comment 2: Figure 1 caption needs to be modified because Figure 1 includes not only the sources of MGO, but also some downstream effects. Also, the location of “AGEs” and “Proteins glycation” may need to be switched because AGE should be one of the consequences of protein glycation.

Reply: We modified Figure 1 caption and content as you suggested.

Comment 3: How do glycated crystallins, albumin, LDL, and ECM lead to ROS and cytotoxicity via glycation? Are they through activation of RAGE, or other AGE receptors? If it is through RAGE, this part of the discussion should be moved to section 2.2; Otherwise, it would be helpful to have a new section 2.3 discussing other downstream receptors of AGEs apart from RAGE.

Reply: The pathways covering ROS formation production from glycated proteins are still lack discussed nowadays. Although RAGE activation can increase cellular ROS level, direct protein glycation leading to ROS production was not reported via RAGE. Therefore, we prefer to keep the original sentences in section 2.1.

Comment 4: It is nice to have some discussion regarding the potential therapeutic application of glyoxalase to detoxify MGO. However, more discussion from a clinical perspective would be helpful for the reader to understand the current stage of drug development targeting MGO. Are there any relevant ongoing clinical trials? If not, what would be the major obstacles to drug development?

Reply: Dear reviewer, only one Glo1 inducer (tRES-HESP) finished phase 1 of clinical trials for the improvement of glucose intolerance and vascular function in overweight and obese patients. We also added information related to other possible therapeutic approaches against DR in section 5 "Future Research Directions and Therapeutic Approaches" (p.10).

Comment 5: Retina is a complex tissue comprising multiple cell types, and they also tightly interact with each other. From the discussions in this manuscript, MGO majorly affects RPE, Muller cells, photoreceptors, and vasculatures. It would be beneficial to briefly summarize the role of MGO on each cell type. Also, are there any tissue engineering models or in vivo models showing how MGO collectively affects them? The following reviews may provide useful context regarding the current retina coculture models:

Marcos LF, Wilson SL, Roach P. Tissue engineering of the retina: from organoids to microfluidic chips. J Tissue Eng. 2021 Dec 10;12:20417314211059876. doi: 10.1177/20417314211059876. PMID: 34917332; PMCID: PMC8669127.

Wu A, Lu R, Lee E. Tissue engineering in age-related macular degeneration: a mini-review. J Biol Eng. 2022 May 16;16(1):11. doi: 10.1186/s13036-022-00291-y. PMID: 35578246; PMCID: PMC9109377.

Reply: The retina is a complex tissue containing various cell types, so their interactions play a sufficient role in disease progression. We found your comment very helpful in improving our review. We added section 5. “Future research directions and therapeutic approaches” (p.10) where discussed this issue.

We would like to thank the referee again for taking the time to review our manuscript.

Reviewer 2 Report

Comments and Suggestions for Authors

The manuscript is interesting. However, several points could be addressed to increase the impact of the work.

Cell types are not well defined. RPE and microglia's evidence is presented. However, other cell types are not discussed. The retina contains various cell types. It is limited to present the work by saying just "retinal cells" in the manuscript. It is important to present cell type-specific explanations with categorization.

AMD and DR's research papers are mixed. However, the current manuscript's focus is DR. Therefore, the AMD part should be removed. 

DR's vasculature should be well summarized. This is one of the most important aspects to damage the retina. VEGF is not the only factor to induce the vasculature injury. 

In this manuscript, the time points of the disease progression are missing. Its aspect is required to list the evidence of the research paper works.

Not only conclusion but also future experimental direction might be needed to be discussed regarding methylglyoxal's roles in DR.

Comments on the Quality of English Language

Grammar should be re-checked during the revision.

Author Response

Dear Reviewer,

We would like to thank the referee for reviewing our manuscript and providing helpful comments. Please find the detailed responses below and the corresponding revisions highlighted changes in the re-submitted files.

Comment 1:Cell types are not well defined. RPE and microglia's evidence is presented. However, other cell types are not discussed. The retina contains various cell types. It is limited to present the work by saying just "retinal cells" in the manuscript. It is important to present cell type-specific explanations with categorization.

Reply: We changed the "retinal cells" to indicate specific cell types motioned in the text. We also added a Table 1 “MGO’s effect on different retinal cell types” summarizing cell type-specific MGO effects.

Comment 2: AMD and DR's research papers are mixed. However, the current manuscript's focus is DR. Therefore, the AMD part should be removed.

Reply: We agree with this comment. As you suggested we deleted the AMD data from our review.

Comment 3: DR's vasculature should be well summarized. This is one of the most important aspects to damage the retina. VEGF is not the only factor to induce the vasculature injury.

Reply: We summarized the pathways involved in neovascularization in section 3.3. “Inflammation and Microglia Activation” (p.5) as you suggested.

Comment 4: In this manuscript, the time points of the disease progression are missing. Its aspect is required to list the evidence of the research paper works.

Reply: We added information about the stages of DR progression in section 1. “Sources of Methylglyoxal and Its Association with Diabetic Retinopathy” (p.1).

Comment 5: Not only conclusion but also future experimental direction might be needed to be discussed regarding methylglyoxal's roles in DR.

Reply: We found your comment very helpful in improving our review. We added part 5. “Future research directions and therapeutic approaches” (p.10) where discussed possible future ways for the DR and MGO researches and targets for DR treatment.

We would like to thank the referee again for taking the time to review our manuscript.

Round 2

Reviewer 2 Report

Comments and Suggestions for Authors

The raised comments are partly resolved.

In the revised manuscript's neovascularization part, aspects on hypoxia-inducible factor activation or inhibition in this disease are missing. As this is an important matter, discussing it is highly recommended.

Author Response

Dear Reviewer, We would like to appreciate your helpful comments and time to review our manuscript. We have addressed your comment:

  1. In the revised manuscript's neovascularization part, aspects on hypoxia-inducible factor activation or inhibition in this disease are missing. As this is an important matter, discussing it is highly recommended.

Reply: Dear Reviewer, as you suggested we added information about the role of hypoxia-inducible factor in the retina's neovascularization to section 3.3 "Inflammation and Microglia Activation" (p.6). Please find highlighted changes in the re-submitted files.

We would like to thank the referee again for taking the time to review our manuscript.